# Disparities and Determinants of Online Medical Record Access among Cancer Survivors

**DOI:** 10.3390/healthcare12161569

**Published:** 2024-08-08

**Authors:** Safa Elkefi

**Affiliations:** Columbia University Irving Medical Center, New York, NY 10032, USA; se2649@cumc.columbia.edu

**Keywords:** online medical records, cancer survivors, digital health, engagement, health disparity

## Abstract

Access to online medical records (OMRs) can help enhance cancer patient engagement and improve their health outcomes. This study investigates disparities in OMR access among cancer survivors and examines the association between OMR access and health perceptions. We conducted a cross-sectional analysis using data from the National Cancer Institute’s Health Information National Trends Survey (HINTS) from 2017 to 2022. The sample included 4713 cancer survivors. We employed regression analysis to assess the associations between the different factors. Overall, 18.78% of participants accessed their OMRs once or twice, while 36.69% accessed them three times or more. Gender minority groups (β = −0.0038, *p* = 0.01), older adults (β = −0.1126, *p* < 0.001), and racial minority groups (β = −0.059, *p* < 0.001) were less likely to access their OMRs. Additionally, higher education levels (β = 0.274, *p* < 0.001), insurance coverage (β = 0.365, *p* < 0.001), and higher incomes (β = 0.115, *p* < 0.001) were associated with increased OMR access. Positive health perceptions were significantly associated with OMR usage, including perceived good health (β = 0.148, *p* < 0.001), quality of care (β = 0.15, *p* = 0.026), and self-efficacy (β = 0.178, *p* = 0.002). Disparities in OMR access among cancer survivors are influenced by socio-economic factors and health perceptions. Interventions targeting vulnerable groups, enhancing digital health literacy, and improving health perceptions could promote equitable OMR usage.

## 1. Introduction

The number of cancer survivors continues to increase in the United States due to the growth and aging of the population, as well as advances in early detection and treatment [1]. As of January 2022, more than 18 million American cancer survivors were alive [1]. This represents approximately 5.4% of the population. This number is projected to increase by 24.4% to 22.5 million by 2032 [1]. For these cancer survivors, complications and long-term consequences related to the diagnosis and treatment of cancer can carry a substantial burden [2,3]. They face several challenges related to the surveillance of cancer treatment and the possibility of recurrence, monitoring of long-term side effects, maintenance of general health, and management of social and psychological facets of recovery, including rehabilitation, adjustment, and reintegration into normal daily life [3,4,5,6].

Considering the extensive amount of information that survivors need to handle, resorting to health information technology, and specifically online medical records, remains an important component of healthcare reform, offering more convenient access to care [7,8,9]. The Institute of Medicine report Crossing the Quality Chasm: A New Health System for the 21st Century posited electronic patient–physician messaging as a promising technology to improve the quality and efficiency of healthcare [7]. Studies suggest that providing patients with online access to their medical records may reduce the need for face-to-face contact [8,10,11]. The presumption is that if patients could look up health information such as their test results, request prescription refills, schedule appointments, and send secure e-mail to clinicians, then their use of clinical in-person and telephone calls may decrease [8,10,11]. Engaging patients and their relatives to play an active role in their healthcare process is a critical element of patient-centered care, yet patients are an underused resource in the healthcare system [12]. Although it is beneficial for cancer patients, accessing online medical records is still limited because of many reasons. Some cancer patients do not prefer the disclosure of bad news online, prefer communicating with doctors directly, and find electronic medical records overwhelming [13]. Some studies have also found that younger adults were more likely to access the records than older adults [14].

It is important to investigate the differences in the utilization of electronic medical records and the factors associated with it. However, most of the research on the factors associated with the utilization of electronic medical records is focused on the general population. This study bridges the gap by exploring (1) disparities in access to electronic health records among cancer patients and (2) the association between this access and health perceptions.

## 2. Materials and Methods

### 2.1. Data and Participants

We used a cross-sectional descriptive survey design to address the study questions. Secondary data were obtained from the National Cancer Institutes’ Health Information National Trends Survey (HINTS). For this study, we used 6 data cycles from 2017 to 2022, including the SEER dataset (Surveillance, Epidemiology, and End Results) for cancer survivors. Only patients with a history of cancer were selected. The HINTS database is a nationally representative survey of the US non-institutionalized adult population [15]. The purpose of the survey is to investigate the American population’s needs in terms of health-related information and their access to care [16,17]. The sampling strategy for the HINTS consists of a two-stage design. In the first stage, a stratified sample of addresses is selected from a file of residential addresses [18]. In the second stage, one adult is selected within each sampled household. Respondents are offered the choice to respond online or through a paper survey [18]. Both modes of the survey (paper and online) were offered in English or Spanish, as per the National Cancer Institute’s rules [18]. All groups received a USD 2 pre-paid monetary incentive to encourage participation [18]. Respondents were offered a bonus incentive to complete the survey online. Returned surveys were reviewed for completion and duplication (more than one questionnaire returned from the same household) to ensure they were eligible for inclusion in the final dataset [18]. The response rate for the survey was 28.1%. Around 41% of the data came from the paper group, and 59% came from web surveys. Further details on the survey design and sampling strategies are published elsewhere [15]. 

### 2.2. Measures

The framework summarizing the variables of the study is detailed in Figure 1.

#### 2.2.1. Dependent Variables

Accessing online medical records: for this variable, we used the question, “How many times did you access your online medical record in the last 12 months?” Responses were “never”, “1 to 2 times”, and “3 times or more”.

#### 2.2.2. Independent Variables

Socio-economic factors: Variables considered included birth gender, sexual orientation, age, education, insurance, and income. Birth gender had “female” and “male” as options. Sexual orientation answers varied between “straight”, “gay or lesbian”, “bisexual”, and “other”. Age groups were “18–34”, “35–49”, “50–64”, “65–74”, and “75 or more”. The education levels considered were “less than high school degree”, “high school graduate”, “some college”, and “college graduate”. For insurance, respondents were asked whether or not they hold any; “yes” or “no”. Income ranges in USD were “0–34,999”, “35,000–99,999”, and “100,000 or more”.

Quality of care: Participants were asked about the quality of care they received in the last 12 months: “Overall, how would you rate the quality of health care you received in the past 12 months?” Responses were on a 4-point Likert scale that we dichotomized into “good” and “bad” for interpretation purposes. 

General health: For the health perception, participants were asked, “In general, would you say your health is good or bad?” with the two options “good” and “bad”.

Self-efficacy: for this variable, we used the question, “Overall, how confident are you about your ability to take good care of your health?” with answers “confident” and “not confident”.

### 2.3. Data Analysis

We merged the HINTS data cycles and selected only cancer survivors. A multiple imputation method was used to handle missing data. Multiple imputations are useful for handling missing data problems and accounting for imputation uncertainty [19,20]. The K-nearest-neighbor algorithm method was used to replace the missing values with the nearest neighbor estimated values [20]. Then, the analyses were conducted. Regression analysis was conducted to assess the association between the independent and dependent variables. A *p*-value of <0.05 was considered significant. Analyses were performed using Python software, version 3.8, using complex survey design procedures (e.g., researchpy, numpy, pandas, statsmodels, semopy, sklearn).

## 3. Results

### 3.1. Characteristics of the Sample

Our study included (N = 4713) cancer survivors from the different HINTS cycles. Most of them were females (57.20%), straight (94.42%), White (74.73%), and college graduates (46.06%). Most of the participants had insurance (98.30%) and had a medium income ranging between 35,000 and 99,999 (44.32%). Table 1 summarizes the socio-economic distribution of the participants.

### 3.2. Online Medical Records Usage among Cancer Survivors

Overall, 18.78% of the participants used the online medical records once or twice only, and 36.69% of them used them three times or more. As shown in Table 2, accessing online medical records was significantly associated with all the socio-economic characteristics but birth gender.

Gender minority groups, older adults, and racial minorities were less likely to access their medical records online (sexual orientation: β = −0.0038, *p* = 0.01; age: β = −0.1126, *p* < 0.001; race: β = −0.059, *p* < 0.001). Higher education levels, insurance, and higher incomes were associated with more access to online medical records (education: β = 0.274, *p* < 0.001; insurance: β = 0.365, *p* < 0.001; income: β = 0.115, *p* < 0.001).

### 3.3. Impact of Health Perceptions on Access to Online Medical Records

As shown in Table 3, we found that cancer patients’ health perceptions were positively significantly associated with access to online medical records. For instance, cancer survivors who thought that they had good health (β = 0.148, *p* < 0.001), those who thought that they received a good quality of care (β = 0.15, *p* = 0.026), and those who thought that they could take care of themselves (β = 0.178, *p* = 0.002) were more likely to access their medical records online.

## 4. Discussion

Cancer survivors’ access to their online medical records (OMRs) serves as one of the cornerstones in the efforts to increase patient engagement and improve cancer care outcomes [13]. This study investigates disparities in cancer survivors’ access to their online medical records and the association between this access and health perceptions. It included N = 4713 cancer survivors, of whom 18.78% used the online medical records once or twice only, and 36.69% used them three times or more. This finding is consistent with previous studies that showed only 25% of cancer patients were persistent users of OMRs, and 35% never accessed the portals [21]. Previous research has also shown that cancer patients are less likely to access their medical records online compared to people without a history of cancer [14,22].

Our findings highlight an association between age and OMR usage, with older adults being less interested, similar to the findings of Franklin et al. [23]. Although previous studies have shown no association between OMR access and race and gender, we found that racial and gender minority groups were less likely to access OMRs compared to White people and heterosexuals, respectively [21]. 

Additionally, we found that higher education levels, insurance, and higher incomes were associated with more access to online medical records. Although no study has investigated the association between these factors and the use of OMRs among cancer patients, it is noteworthy that research has shown that a greater use of digital health was observed among these groups [24,25].

Finally, we found that cancer patients’ health perceptions were positively significantly associated with access to online medical records. For instance, cancer survivors who thought that they had good health, those who thought that they received a good quality of care, and those who thought that they could take care of themselves were more likely to access their medical records online. Previous studies have confirmed that cancer patients who think that they are not vulnerable and who have a good perception of their health conditions are more eager to engage in healthy behaviors [26,27]. This may explain why they would want to adopt technology to help them gain more autonomy and self-acceptance [28].

### 4.1. Practical Implications

These findings have significant clinical implications for improving patient engagement, personalized care, and overall health outcomes among cancer survivors.

First, given that gender minorities, older adults, and racial minorities are less likely to access OMRs, targeted interventions should be adopted to enlarge the reach of these technologies. Clinicians should be aware of these disparities and offer personalized tutorials on using portals, follow up regularly on OMR usage, and address the specific barriers that patients face. 

Additionally, we found that socio-economic factors such as education, income, and insurance can play a crucial role in digital health engagement. Promoting more portal usage may require coordination with social services to provide the necessary support, such as financial assistance, for internet access or digital literacy programs. 

Moreover, the significant association between positive health perceptions and OMR usage indicates that patients who feel healthier and more in control are more likely to engage with their medical records. Empowering cancer patients and supporting their confidence in their health is important. Encouraging them also to access their health records online could help foster a sense of empowerment and proactive health management. Clinicians can use OMRs as tools to provide personalized care, tailoring their approach based on the patient data available in these records. By encouraging the regular use of OMRs, patients can maintain up-to-date information on their health, which can aid in more accurate and timely clinical decision making. In this study, we only considered insurance coverage (present or not present). In future studies, it would be more informative to consider the type of coverage the patients have for better differentiation in the outcomes. 

### 4.2. Limitations

This study has some limitations that are worth acknowledging. First, its cross-sectional design may limit the ability to establish causality between the variables studied. Second, the reliance on self-reported data from the HINTS may introduce recall bias or social desirability bias. Participants might overestimate or underestimate their usage of OMRs and their health perceptions. Additionally, while the study identifies socioeconomic disparities and associations with health perceptions, it does not delve deeply into the specific barriers and facilitators of OMR access among different subgroups. More qualitative research might be needed to understand the underlying reasons behind the observed patterns. Finally, the absence of longitudinal data prevents this study from tracking changes in OMR usage and health perceptions over time among the same individuals, which could provide more insights into trends and causative factors.

## 5. Conclusions

Our findings have important implications for healthcare providers and policymakers. To bridge the gap in OMR access, tailored interventions should be developed to support vulnerable groups, including personalized tutorials, regular follow-ups, and addressing specific barriers to OMR usage. Enhancing digital health literacy through educational programs and support services is crucial. Additionally, improving patients’ overall health perceptions by providing comprehensive, high-quality care can further encourage OMR engagement. By addressing these disparities and promoting equitable access to OMRs, healthcare systems can enhance patient engagement, leading to better health outcomes and more personalized cancer care. Future research should explore the underlying barriers to OMR access and develop strategies to overcome these challenges, ensuring that all cancer survivors have the opportunity to benefit from digital health technologies.

## Figures and Tables

**Figure 1 healthcare-12-01569-f001:**
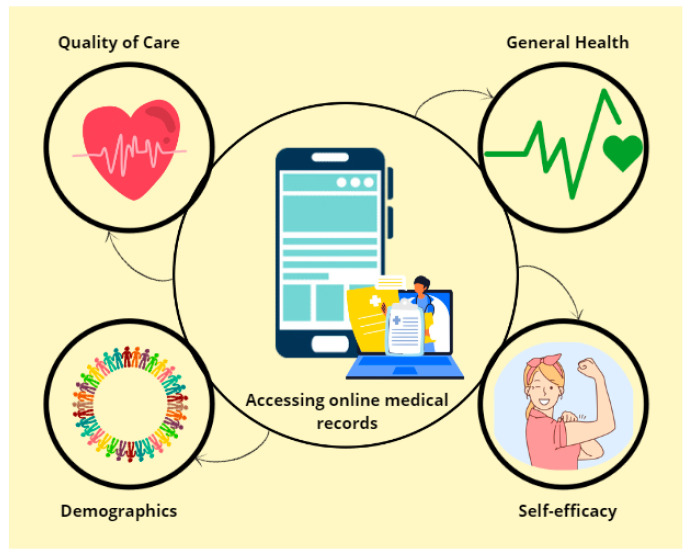
The framework of the study of the possible predicting factors of online medical records access.

**Table 1 healthcare-12-01569-t001:** Characteristics of the participants.

Variable	Variable Group	N	%
	All	4713	100%
Birth gender	Male	2017	42.80%
	Female	2696	57.20%
Sexual orientation	Straight	4450	94.42%
	Gay or lesbian	99	2.10%
	Bisexual	46	0.98%
	Other	118	2.50%
Age	18–34	53	1.12%
	35–49	256	5.43%
	50–64	1205	25.57%
	65–74	1649	34.99%
	75 or more	1550	32.89%
Education	<High school	243	5.16%
	High school grad	860	18.25%
	Some college	1439	30.53%
	College grad or more	2171	46.06%
Race	White	3522	74.73%
	Black	563	11.95%
	Hispanic	396	8.40%
	Asian	126	2.67%
	Other	106	2.25%
Insurance	No	80	1.70%
	Yes	4633	98.30%
Income ranges	0–34,999	1405	29.81%
	35k–99,999	2089	44.32%
	100k or more	1219	25.86%

**Table 2 healthcare-12-01569-t002:** Accessing online medical records among the different participants.

	Access Online Record	Never	1–2 Times	3 Times or More	Coeff	*p*-Value
	All	44.54%	18.78%	36.69%		
Birth gender	Male	44.62%	17.95%	37.43%	−0.0054	0.873
	Female	44.66%	18.62%	36.72%		
Sexual orientation	Straight	44.20%	18.58%	37.21%	−0.0038	0.01
	Gay or lesbian	35.35%	17.17%	47.47%		
	Bisexual	36.96%	15.22%	47.83%		
	Other	72.03%	11.02%	16.95%		
Age	18–34	43.40%	20.75%	35.85%	−0.1126	<0.001
	35–49	36.33%	17.97%	45.70%		
	50–64	37.18%	21.00%	41.83%		
	65–74	43.48%	18.31%	38.20%		
	75 or more	52.77%	17.61%	29.61%		
Education	<High school	74.90%	14.40%	10.70%	0.2741	<0.001
	High school grad	61.51%	17.56%	20.93%		
	Some college	46.77%	19.46%	33.77%		
	College grad or more	33.16%	18.33%	48.50%		
Race	White	41.20%	18.94%	39.86%	−0.0599	<0.001
	Black	61.99%	15.28%	22.74%		
	Hispanic	53.54%	18.69%	27.78%		
	Asian	34.92%	13.49%	51.59%		
	Other	45.28%	18.87%	35.85%		
Insurance	No	61.25%	21.25%	17.50%	0.365	<0.001
	Yes	44.36%	18.28%	37.36%		
Income ranges	0–34,999	61.57%	15.73%	22.70%	0.01151	<0.001
	35k–99,999	41.89%	19.67%	38.44%		
	100k or more	29.86%	19.03%	51.11%		

**Table 3 healthcare-12-01569-t003:** Association between health and care perceptions and online medical records access.

Variable Group	Accessing Online Medical Records	
	Coeff	*p*-Value
Quality Care	0.1502	0.026
general health	0.1484	<0.001
Self-efficacy	0.1778	0.002

## Data Availability

Data are available online on the HINTS data section of the NCI website.

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
