# Peer review of "Disparities and Determinants of Online Medical Record Access among Cancer Survivors"

_healthcare, 2024, doi:10.3390/healthcare12161569_

Round 1

Reviewer 1 Report

Comments and Suggestions for Authors

The authors conducted a cross-sectional analysis on data gathered from the National Cancer Institute's Health Information National Trends Survey (HINTS) from 2017 to 2022. The sample included 4,713 cancer survivors

The study investigates demographic disparities in online medical records (OMR) access among cancer survivors and examines the association between OMR access and health perceptions. 

Although the methodology is rather trivial and the classification too simple, there are some interesting outcomes that are worthy to be published. The authors must elaborate on the following issues that are reported in detail.

Introduction Section

1.       Page 1 line 28 : “As of January 2022, more than 18 million Americans with a history of cancer were alive” Please can the author provide more information ? From 2022 to when? What types of cancer?

2.       Page 1 lines 41-42:  “Studies suggest…” which studies? In which countries, please provide a ref and more information for the readers.

3.       Page 2 lines 51-52: “Some studies” again, please provide refs

Materials and Methods section

4.       Page 2, line 66: It would be interesting if the author could report the types of cancer? even the stage of cancer patients, at the time of the study?

5.       Page 2 line 74: Can the author elaborate more about why the second language was Spanish?

6.       Page 3 lines 85-87: It is not clear to me whether the responses were open-ended or whether there was a specific scale? And if there was a specific scale, how was it chosen?

7.       Page 3 lines 98-101: The question is how the patients who were neither satisfied nor dissatisfied with the services were tracked? Some questions about the general health and Self-efficacy.

Results section

8.       Page 3 lines 121-122 I would suggest mentioning that number refers to dollars!

9.       Suggestion: It would be very useful for the authors to investigate by type of cancer and then compare these results between them.

Discussion section

10.    Page 6 lines 171-174please report more info about these studies, in which countries these studies were performed?

11.     Page 6 lines 173-174. Did the people without cancer history have another disease? Or the comparison is with healthy people?

12.    Page 7 lines 177-179: what the authors state about the differences in results between these studies?

Author Response

Introduction Section

  • Page 1 line 28 : “As of January 2022, more than 18 million Americans with a history of cancer were alive”Please can the author provide more information ? From 2022 to when? What types of cancer?

Response: Please note that "as of" means by the statistics of that date. The stats include all types of cancer. This was clarified in the text.

  • Page 1 lines 41-42:  “Studies suggest…” which studies? In which countries, please provide a ref and more information for the readers.

Response: We added citations that support the information provided. Thank you for the comment.

  • Page 2 lines 51-52: “Some studies” again, please provide refs

Response: We added citations that support the information provided. Thank you for the comment.

Materials and Methods section

  • Page 2, line 66: It would be interesting if the author could report the types of cancer? even the stage of cancer patients, at the time of the study?

Response: We thank you for the comment. Although it is relevant, we did not collect that information as part of the data collection and analysis. We will include that in the imitation section and future studies.

  • Page 2 line 74:Can the author elaborate more about why the second language was Spanish?

Response: We thank you for the comment. Spanish is the second most spoken language in the US which is why most of the data collected publicly by the NIH as in both Spanish and English. 

  • Page 3 lines 85-87: It is not clear to me whether the responses were open-ended or whether there was a specific scale? And if there was a specific scale, how was it chosen?

Response: We thank you for the comment. Please note that the information was clearly  stated in the article:

Example: 

Accessing online medical records: for this variable, we used the question, "How many times did you access your online medical record in the last 12 months?" Responses were "never", "1 to 2 times", and "3 times or more".

That shows that the responses were on scales and not open-ended.

Same goes for the other questions. All of the scales were clearly declared:

"Demographics: Demographic variables considered included birth-gender, sexual orientation, age, education, insurance, and income. Birth gender had "female" and "male" as options. Sexual orientation answers varied between "straight", "gay or lesbian", "bisexual", and "other". Age groups were "18-34", "35-49", "50-64", "65-74", and "75 or more". Education levels considered were "less than high school degree", "high school graduate", "some college", and "college graduate". For insurance, respondents were asked whether or not they hold any; "yes" or "no". Income ranges were "0-34,999", "35,000-99,999", and "100,000 or more".

Quality of care: Participants were asked about the quality of care they received in the last 12 months: "Overall, how would you rate the quality of health care you received in the past 12 months?" Responses were on a 4-likert scale that we dichotomized into "good" and "bad" for interpretation purposes.

General health: For the health perception, participants were asked, "In general, would you say your health is good or bad?" with the two options "good" and "bad".

Self-efficacy: for this variable, we used the question, "Overall, how confident are you about your ability to take good care of your health?" with answers "confident" and "not confident"."

  • Page 3 lines 98-101: The question is how the patients who were neither satisfied nor dissatisfied with the services were tracked? Some questions about the general health and Self-efficacy.

Response: We thank you for the comment. We did not design the survey; this is a secondary data analysis. The NIH designed the initial survey that way. We agree that it could be a limitation, not to capture neutral answers.

Results section

  • Page 3 lines 121-122 I would suggest mentioning that number refers to dollars!

Response: We thank you for the comment. We corrected that:

Income ranges in dollars were "0-34,999", "35,000-99,999", and "100,000 or more".

  • Suggestion: It would be very useful for the authors to investigate by type of cancer and then compare these results between them.

Response: We thank you for the comment. As mentioned earlier. We don't have access to that type of information. For this study, we are also interested in the cancer diagnosis independently of the type.

Discussion section

  • Page 6 lines 171-174, please report more info about these studies, in which countries these studies were performed?

Response: We thank you for the comment. This information about where the studies took place are not relevant to the scope of the article. We also do not think it would add any further information to the point being discussed.

  • Page 6 lines 173-174. Did the people without a cancer history have another disease? Or the comparison is with healthy people?

Response: We thank you for the comment. We don't think this information is relevant to this scope. We are not comparing to any groups. We are assessing the access of online medical records among cancer patients. That's it.

  • Page 7 lines 177-179: what the authors state about the differences in results between these studies?

Response: We thank you for the comment. No difference was noted. All of those studies found the same result that we mentioned there. We don't think it would be relevant to dig deeper in the differences, here. 

Reviewer 2 Report

Comments and Suggestions for Authors

A well-done two-part correlation study with a few exceptions. The first part (demographic factors and access) is presented independently of the second part (healthcare perceptions and access). The discussion, practical limitations and conclusions are consistent with the findings as well as suggestions for future interventions and research. In addition, the results are consistent with what we know about socio-economic disparities in the USA health care system.

There is a slight issue with terminology.  Demographics and socio-economic factors are both used.  My preference is to use “socio-economic” which is a broader term.

The data are very appropriate, and the author describes the essentials of HINTS, as well as the two-stage sampling strategy. I have a few observations in this regard. Mention is made of a control group, Line 75. Please clarify.  

A paper and online survey was used. What is the percentage of respondents for each method?  Please comment on whether these yielded groups which are similar or dissimilar in composition and any effect on the statistical results and external validity.  I do not detect mention of a response rate for an overall response rate yielding a sample of 4,713. How many respondents and what percentage used the Spanish survey version as opposed to the English version. I realize that some of this information is published elsewhere, but it should be included in this paper.

The analysis of the socio-demographic variables, Table 1, and the perceptual variables, Table 3, could benefit from a quantitative measure of association to assess the relative impact of the variables, those which are significant or non-significant. This is helpful for the discussion and practical limitations.

The socio-economic variables are ones that you normally find in studies of this type. From an “insurance” perspective it is often beneficial to distinguish among different coverage types – public including Medicare and Medicaid, and private. Future studies which include this distinction may have important policy implications.

“Quality Care,” “general health,” and “Self-efficacy” are basic variables. In studies of this type I expected at least one follow-up question for each. The analysis would have been more substantive. Can the author explain why this was not done? (Note: Consistency in variable labels is necessary)

This study meets the minimum requirements for a correlational study. Figure 1 does not have a title in the proper format and is slightly misleading to me. While this is a minor point, I do not understand why a combined multivariate analysis was not used. The value and significance of this paper does not achieve its potential. 

Author Response

A well-done two-part correlation study with a few exceptions. The first part (demographic factors and access) is presented independently of the second part (healthcare perceptions and access). The discussion, practical limitations and conclusions are consistent with the findings as well as suggestions for future interventions and research. In addition, the results are consistent with what we know about socio-economic disparities in the USA health care system.

Response: We thank you for your feedback. 

There is a slight issue with terminology.  Demographics and socio-economic factors are both used.  My preference is to use “socio-economic” which is a broader term.

Response: We thank you for your comment. We updated the full paper following this comment.

  • The data are very appropriate, and the author describes the essentials of HINTS, as well as the two-stage sampling strategy. I have a few observations in this regard. Mention is made of a control group, Line 75. Please clarify.  

Response: We thank you for your comment. We corrected that. There is no control group.

  • A paper and online survey was used. What is the percentage of respondents for each method? Please comment on whether these yielded groups which are similar or dissimilar in composition and any effect on the statistical results and external validity. 

Response: We thank you for your comment. Please note that while HINTS is transparent on the way the data is collected. The source of where each instance of the data is coming from is not reported in the Methodology report. The information reported is the same from both groups so they are treated as same data.

  • I do not detect mention of a response rate for an overall response rate yielding a sample of 4,713. How many respondents and what percentage used the Spanish survey version as opposed to the English version. I realize that some of this information is published elsewhere, but it should be included in this paper

Response: We thank you for your comment. The response rate was of 28.1%. Around 41% of the data came from the paper group, and 59% came from Web surveys. We added that to the methods section. As per the Spanish/ English responsiveness, it is not reported. The survey is provided in Spanish but then the answers are coded in English as per the Methodology report. 

  • The analysis of the socio-demographic variables, Table 1, and the perceptual variables, Table 3, could benefit from a quantitative measure of association to assess the relative impact of the variables, those which are significant or non-significant. This is helpful for the discussion and practical limitations.

Response: We thank you for your comment. Please note that while the Table 1 only reports prevalence and rates (N, %), the sociodemographic factors are present in table 2 with statistical significance assessed. Table 3 as well reports P-values measuring the correlation between the variables.

  • The socio-economic variables are ones that you normally find in studies of this type. From an “insurance” perspective it is often beneficial to distinguish among different coverage types – public including Medicare and Medicaid, and private. Future studies which include this distinction may have important policy implications.

Response: We thank you for your comment. We added this information to the limitations of the study. 

"In this study, we only considered the insurance coverage (present or not present). In future studies, it would be more informative to consider the type of coverage the patients have for better differentiation in the outcomes. "

  • “Quality Care,” “general health,” and “Self-efficacy” are basic variables. In studies of this type I expected at least one follow-up question for each. The analysis would have been more substantive. Can the author explain why this was not done? (Note: Consistency in variable labels is necessary)

Response: We thank you for your comment. Please note that this is a cross-sectional study. We are conducting a secondary data analysis and not collecting data ourselves. HINTS are considering longitudinal data in their upcoming cycles.

  • This study meets the minimum requirements for a correlational study. Figure 1 does not have a title in the proper format and is slightly misleading to me. While this is a minor point, I do not understand why a combined multivariate analysis was not used. The value and significance of this paper does not achieve its potential. 

Response: We thank you for your comment. We do not agree with you, though. Please refer to tables 2 and 3 for the analysis. Figure 1 is just a conceptual framework of the predicting factors tested. We changed the title for more clarity. Regression analysis was conducted to test the different associations.